# De novo spatiotemporal modelling of cell-type signatures in the developmental human heart using graph convolutional neural networks

Sergio Marco Salas[1], Xiao Yuan[2], Christer Sylven[3], Mats Nilsson[1]*, Carolina Wählby[2]*, Gabriele Partel[2,4,5]*

1 Science for Life Laboratory, Department of Biochemistry and Biophysics, Stockholm University, Solna, Sweden, 2 Department of Information Technology and Science for Life Laboratory Uppsala University, Uppsala, Sweden, 3 Department of Medicine, Karolinska Institutet, Huddinge, Stockholm, Sweden, 4 Laboratory of Multi-omic Integrative Bioinformatics, Department of Human Genetics, KU Leuven, Leuven, Belgium, 5 VIB-KU Leuven Center for Brain & Disease Research, Laboratory of Computational Biology, Department of Human Genetics, Leuven, Belgium

* mats.nilsson@scilifelab.se (MN); carolina.wahlby@it.uu.se (CW); gabriele.partel@kuleuven.be (GP)

**Data Availability Statement:** The code for spage2vec is available as a Jupyter Notebook (written in python) at http://www.doi.org/10.5281/

## Abstract

With the emergence of high throughput single cell techniques, the understanding of the molecular and cellular diversity of mammalian organs have rapidly increased. In order to understand the spatial organization of this diversity, single cell data is often integrated with spatial data to create probabilistic cell maps. However, targeted cell typing approaches relying on existing single cell data achieve incomplete and biased maps that could mask the true diversity present in a tissue slide. Here we applied a *de novo* technique to spatially resolve and characterize cellular diversity of in situ sequencing data during human heart development. We obtained and made accessible well defined spatial cell-type maps of fetal hearts from 4.5 to 9 post conception weeks, not biased by probabilistic cell typing approaches. With our analysis, we could characterize previously unreported molecular diversity within cardiomyocytes and epicardial cells and identified their characteristic expression signatures, comparing them with specific subpopulations found in single cell RNA sequencing datasets. We further characterized the differentiation trajectories of epicardial cells, identifying a clear spatial component on it. All in all, our study provides a novel technique for conducting de novo spatial-temporal analyses in developmental tissue samples and a useful resource for online exploration of cell-type differentiation during heart development at sub-cellular image resolution.

## Author summary

With the boom of spatially-resolved transcriptomics methods, a set of big opportunities appear for researchers to understand better developmental processes without relying on existing single cell RNA sequencing datasets. In context, we applied spage2vec, a graph convolutional neural network model, to explore the cellular diversity present during the human heart development from week 4.5 to week 9. We showed that this type of

zenodo.4030404, while the specific code used to perform spage2vec on the developing human heart ISS data can be found in: https://github.com/wahlby-lab/spage2vec_heart (http://www.doi.org/10.5281/zenodo.6591405). An online TissUUmaps (14) viewer for interactive exploration of the analysis results can be found in: https://tissuumaps.research.it.uu.se/human_heart.html. All the data generated in this study can be downloaded from the TissUUmaps viewer for further exploration.

**Funding:** The project was funded by the European Research Council via ERC Consolidator grant 682810 and Swedish Foundation for Strategic Research (grant BD150008), the Chan Zuckerberg Initiative (SVCF 2017-173964), an advised fund of Silicon Valley Community Foundation; Erling-Persson Family Foundation; Knut and Alice Wallenberg Foundation (KAW 2018.0172) and Vetenskapsrådet [2019-01238]. The funders had no role in study design, data collection and analysis, decision to publish, or preparation of the manuscript.

**Competing interests:** The authors have declared that no competing interests exist.

approaches, which do not rely on cellular segmentation, can be used to identify molecular signatures better than existing cell typing tools that can be used to explore the spatial component of processes such as cellular differentiation. Among the signatures identified, we focused on exploring the molecular diversity within cardiomyocytes and we described the differentiation process of epicardial cells in the context of space. All the molecular signatures detected and their gene expression profiles were also integrated in a useful online tool that can be used by the community for further exploration.

## Introduction

Recent cell atlasing efforts to describe the cellular complexity of human organs, and provide comprehensive maps of their cell types, have been supported by both technological developments and a number of international initiatives [1,2]. One such technological advance is single-cell RNA sequencing (scRNA-seq) [3,4], enabling profiling the transcriptome of tens of thousands of individual cells after tissue dissociation, and thus defining the cell-type composition of a tissue. Single cell RNA sequencing data can be further combined with more recent spatially resolved techniques [5,6] to create organ-wide gene expression atlases that map cell-type distributions and spatial biological programs directly in tissue samples.

With the aim of producing a spatiotemporal gene expression and cell atlas of the developmental human heart, Asp et al. [7] recently combined three different high-throughput technologies for gene expression profiling with immunohistochemical staining. They studied three developmental stages in the first trimester at 4.5–5, 6.5 and 9 post-conception weeks (PCW) using (1) Spatial Transcriptomics [8] (ST) for untargeted spatial gene expression profiling of grid-microdissected tissues, (2) scRNA-seq of 6.5 PCW tissue samples for dissecting cellular heterogeneity at single cell resolution, and (3) in situ sequencing [9] (ISS) to resolve the spatial heterogeneity at subcellular resolution. Finally, probabilistic cell-mapping via pciSeq [10] was applied to achieve single cell level maps of the cell-type distribution. PciSeq both assigns in situ decoded reads to segmented cells and cells to scRNA-seq defined cell-types by a probabilistic approach. The result of the overall study was the first comprehensive spatial transcriptional atlas of the developmental human heart.

Despite this achievement, some important limitations were found in the study, as pointed out also by Phansalkar et al. [11] in a commentary to the paper. One of the main limitations of the probabilistic approach is that preexisting knowledge of the tissue constituent cell-types is required to characterize the spatial cellular heterogeneity. Thus, probabilistic cell typing by in situ sequencing (pciSeq) was only possible for the 6.5 PCW developmental stage where single cell data was available, leaving the cellular diversity in the 4.5–5 and 9 PCW time points unexplored at a single cell level. Additionally, cell-typing methods that depend on priors defined by scRNA-seq, such as pciSeq or Tangram [12], may introduce a strong bias that can limit the possibility to distinguish between rare cell sub-types or sub-states that might not be fully captured by scRNA-seq sampling approaches, thus preventing to resolve the full range of heterogeneity of a sample. Finally, most of the current cell-typing methods rely on their ability to segment out 3D cells from a 2D representation of the tissue, leading to possible misidentification of cells and misclassification of reads.

In order to overcome these limitations, we present a *de novo* spatiotemporal analysis of the Asp et al. ISS data where we model all three developmental stages at the same time. For our analysis we applied a data driven approach, called spage2vec [13], that generates a common representation of the spatial gene expression at the different developmental stages. Spage2vec

represents the spatial gene expression as a graph and applies a graph representation learning technique to create a lower dimensional representation of the data independent from scRNA-seq defined priors. We then used this common data representation to define the identities and the spatiotemporal relationship of cell- and sub-cell-type gene expression signatures across the three developmental stages of the embryonic heart, identifying previously unreported molecular identities within cardiomyocytes as well as in atrial sub-epicardial cells. We provide newly molecularly defined maps of cell-type signatures during embryonic human heart development on an online platform for interactive exploration (https://tissuumaps.research.it.uu.se/human_heart.html)[14], that together with the Asp et al. atlas[7] represents a useful resource for future studies on human heart development.

## Results

### Identification of *de novo* cellular signatures during the early heart development

With the aim of exploring cellular diversity during the human heart development, spage2vec [13] was used to identify cellular expression signatures *de novo* across the three developmental stages (**Fig 1A**) (Materials and Methods). The analysis is based on the locations of genes represented in the gene panel used during ISS, and is thus dependent on the panel's ability to represent heterogeneity in gene expression. A total of 27 clusters with specific cellular expression signatures were found across the three time points of the heart development (**Fig 1A and 1B**). The signatures were grouped in five main classes, and annotated according to their expression patterns, including atrial cardiomyocytes, ventricular cardiomyocytes, fibroblast-like cells/epicardium-derived cells, epicardial cells and neural crest cells (**Fig 1B**). Clusters assigned to the same class were found to have similar molecular and spatial patterns (**Fig 1C and 1D**). Their distribution was also found to be consistent through the different samples and time points analyzed and most of them were found to have a conserved location in the heart between PCW 4.5–5, PCW 6.5 and PCW 9 (**Fig 1D**).

### Spage2vec represents the cellular diversity better than pciSeq

In order to identify common findings and discrepancies with previous results, we compared the cellular identities defined by spage2vec with the ones described in *Asp et al.* [7]. This comparison was limited to the only time point analyzed via probabilistic pciSeq[10]; PCW 6.5 (**S1 Fig**, Materials and Methods). A total of 27 meaningful cellular identities shared across all three time points were defined, in contrast to the 15 cell types defined from scRNA-seq data and assigned in situ via pciSeq in *Asp et al.* Most of these additional clusters capture a previously undescribed diversity within cardiomyocytes (**S1 Fig**), while other cell types such as endothelial cells or fibroblast-like cells present a one-to-one correspondence. This is also observed when comparing the expression signatures of the spage2vec clusters and the cell types described in the single cell RNA sequencing dataset from *Asp et al.* (**S1 Fig**). Regarding the spatial location of the clusters, both methods agreed on the location of some clusters such as epicardial cells and, to a lesser extent, capillary endothelial cells (**S2 Fig**). However, more striking differences were observed when comparing the location of other cell types, such as the clusters with a fibroblast-like expression signature, where spage2vec clusters present a more specific spatial distribution through the tissue. (**S1 Fig**).

### Individual time point analysis uncovers time-point specific expression signatures

To investigate whether the larger PCW 9, samples with a higher number of cells, could be driving the clustering results, leading to a misclassification of cells in the earlier and smaller tissue

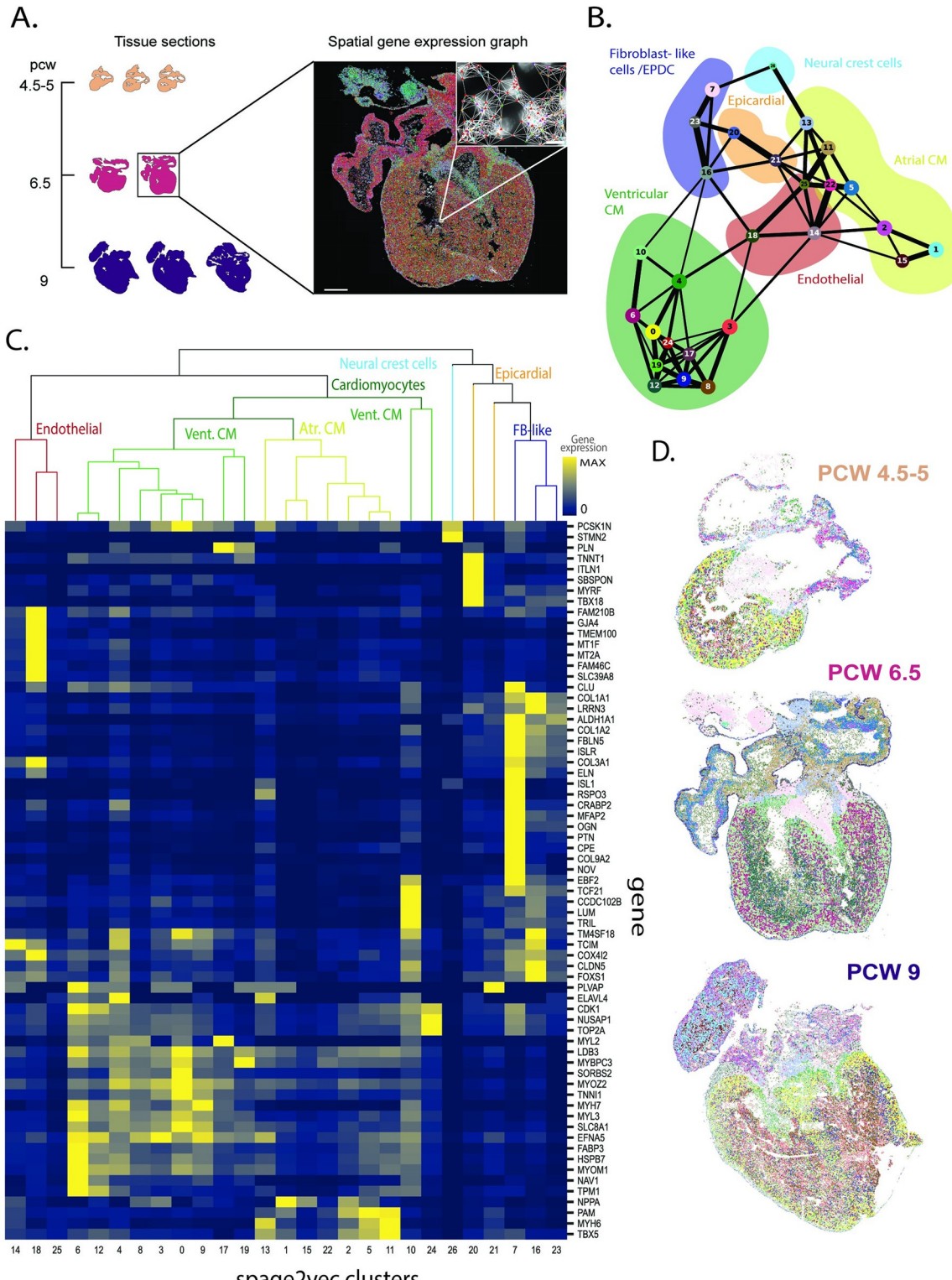

**Fig 1. Overview of the Spage2vec approach to characterize the human developmental heart. A.** Spage2vec constructs a graph from the spatial gene expression of the tissue samples and projects spatial markers in a common latent space. Scale bars: 1 mm, cutout 15 μm. **B.** PAGA plot representing the different expression profiles defined using spage2vec. Background colors represent main cell classes manually annotated based on cluster expression profiles. **C.** Heatmap showing the mean expression of each gene (i.e. expression profile) in the clusters defined by spage2vec, along with suggested cell classes, color coded as in Fig 1C. Expression is normalized by gene **D.**

Spatial maps of the different expression profiles defined using spage2vec in three sections (color coded as Fig 1B), one from each time point. For interactive multi-resolution viewing: https://tissuumaps.research.it.uu.se/human_heart.html.

samples, individual clustering was performed separately on each of the time points (Materials and Methods). A total of 94 clusters were found, including 34 in PCW 4.5–5 and 30 both in PCW 6.5 and PCW 9 (**Fig 2**). Overall, the clusters found in the different time points present a similar distribution in the latent space for all three time points (**S3 Fig**). In addition, most clusters found in specific time points recapitulated molecular and spatial signatures found when analyzing all time points together (**S3 Fig**). The clear correspondence between both analyses shows that the diversity found at different time points was not driven by any of the samples individually (**S4 Fig**).

One important strength of the *de novo* analysis presented here is its ability to resolve the cellular heterogeneity at a higher resolution compared to the scRNA-seq data driven analyses, finding a larger number of clusters with distinct and consistent spatial distribution across the different samples, despite that the molecular information per cell is much lower than in the scRNAseq. One reason for this is the vastly larger number of cells analyzed in the ISS experiment (20.920 cells), compared to the scRNA seq experiment (3.777 cells). In order to assess whether this spatially defined diversity could also be found in the scRNA-seq dataset, the molecular signatures from the intermediate time point samples (PCW 6.5) and its corresponding scRNA-seq dataset were integrated using SpaGE [15] (Method). As observed in Fig 2B, several molecular signatures matched specific subpopulations in the single cell dataset at week 6.5. To compare the differences between the subpopulations, we identified their most differentially expressed genes (**Figs 3A and S5**) (Materials and Methods) and assessed their gene ontology (GO) characteristics using scRNA-seq and compared their spatial locations in the tissue (**S7 Fig**).

## Cardiomyocytes present a previously uncaptured cellular diversity

The developing heart's cardiomyocytes provides a clear example where spage2vec clusters identify high molecular diversity. Three different molecular signatures were found within atrial cardiomyocytes and a total of five molecular signatures were found within ventricular cardiomyocytes, three of them having supporting scRNA-seq data (**Fig 2B**). Top differentially expressed genes (DEG) between the different subtypes were found to be expressed only in cardiomyocytes, presenting in scRNAseq the same coexpression patterns as in the ISS dataset, and suggesting that the diversity identified within this group of cells is not caused by the influence of neighboring cells (**S6 Fig**). These newly identified clusters also present a better-defined region-specific location compared to analogous pciSeq cell-type maps in Asp et al., where some atrial cells are misplaced in the ventricles and vice versa (**S1 Fig**). Within ventricular cardiomyocytes, the five different spage2vec clusters presented unique expression patterns and spatial distributions from the periphery to the interior of the heart. However, not all the clusters were aligned with corresponding cell subpopulations from scRNA-seq data integration (Method). While clusters 4, 8 and 15 aligned within both ventricular and MYOZ2-enriched cardiomyocytes, cluster 16 and 24 presented a very weak alignment within the cells described in the scRNA-seq analysis (**Fig 2B**). Indeed, we related pcw6.5–16 with the presence of some endothelial markers the neighboring cells, suggesting a mixed signature (**S7 Fig**). We found cluster 4 to be located within the ventricular wall, presenting a high expression of MYH7 and characteristics of trabecular myocardium. In contrast, cluster 8 had a peripheral location and was also found to have a strong expression of MYH7 consonant with outer, compact myocardium (**Fig 3A**). Both molecular signatures had GO characteristics of contracting ventricular

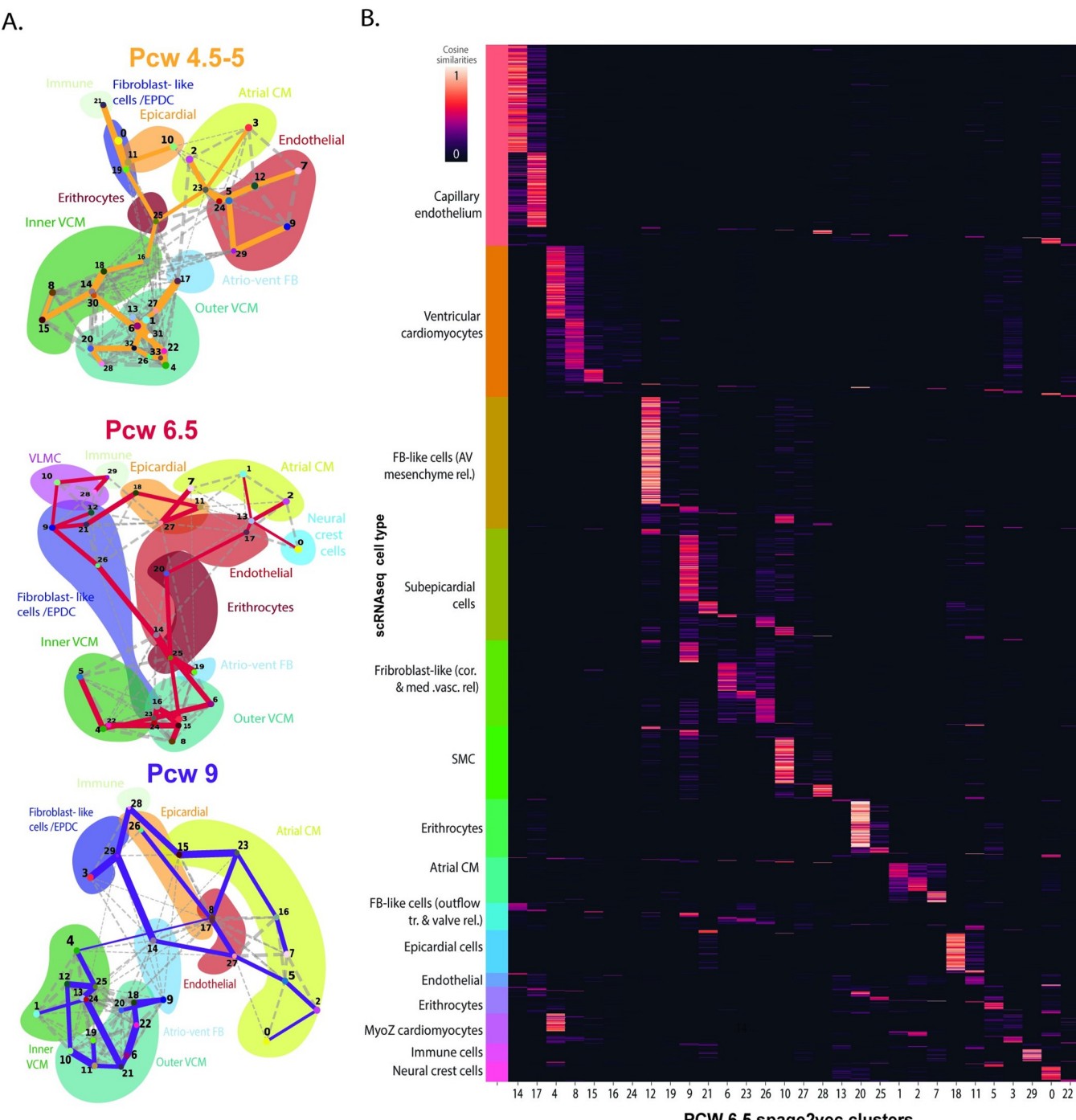

**Fig 2. Analysis of individual developmental stages and correspondence with single cell data. A.** PAGA plots of the clusters found in each time point specific analysis. Clusters from each time point are represented in a PAGA plot, including 4.5–5 PCW (top), 6.5 PCW (middle) and 9 PCW (bottom). Background colors represent the main cell types found in the dataset. **B.** Heatmap representing correspondence in terms of cosine similarities between scRNA-seq data and spage2vec clusters from PCW 6.5 (Materials and Methods), normalized by row, therefore ranging from 0 to 1. Cells from scRNA-seq dataset (rows) are sorted based on their cell type in order to facilitate the interpretation.

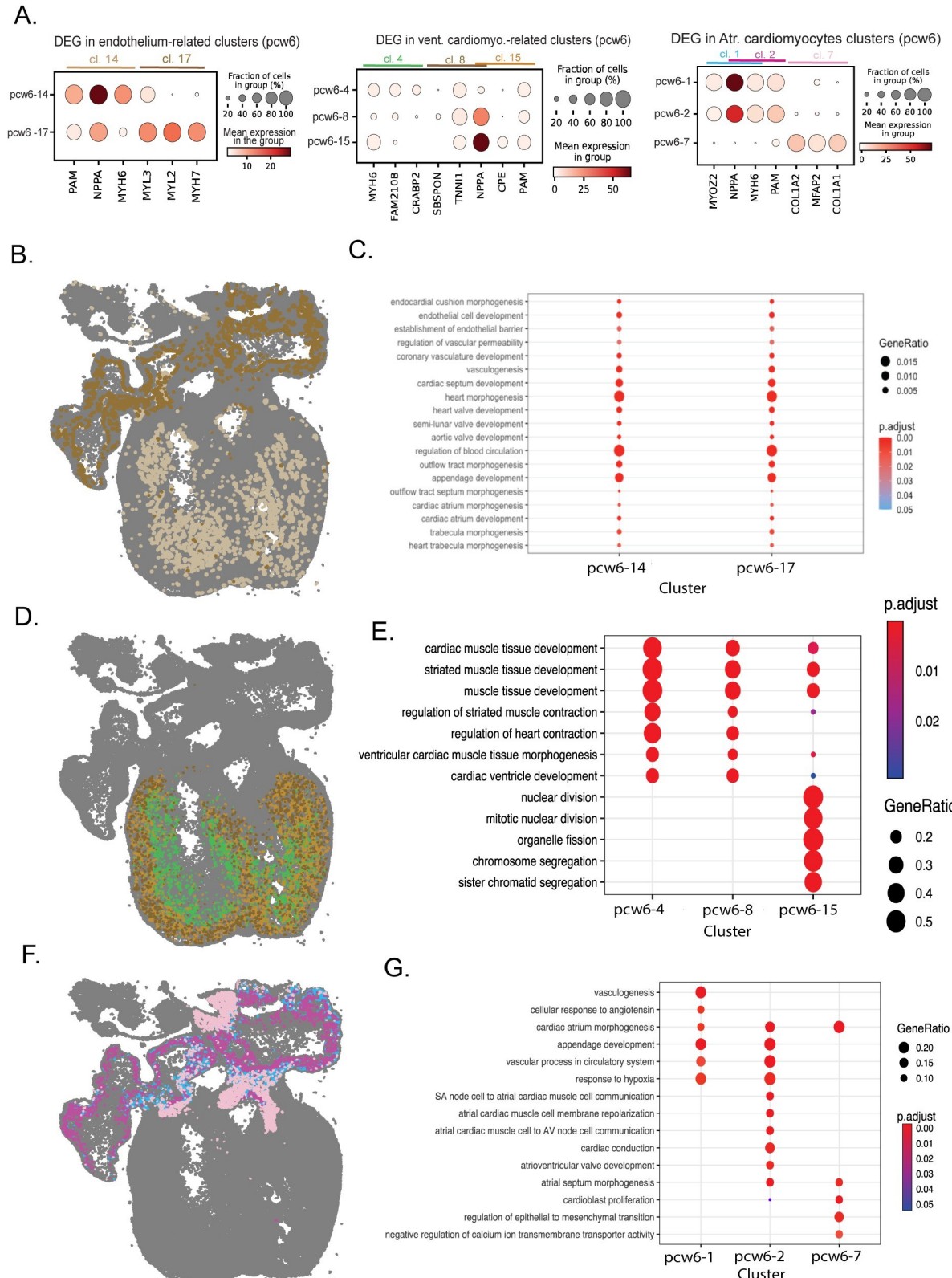

**Fig 3. Exploration of molecular signatures identified within cardiomyocytes and endothelial cells. A.** Dot plots representing the expression of the 3 most differentially expressed genes between each of the clusters (cl) related with endothelial cells (cluster 14 and 17)(left),

ventricular (cluster 4, 8 and 15)(middle) and atrial (cluster 1,2 and 7)(right) and cardiomyocytes. Expression is shown in the endothelial and cardiomyocyte related clusters linked to specific populations within the scRNA-seq dataset from Asp.et al.[7] **B.** Spatial maps highlighting the reads assigned to the clusters related with endothelial cells and cardiomyocytes (right), with gene ontology enrichment of biological processes for the top 15 most differentially expressed genes of each cluster (left). Color codes as in A.

muscle although these GO terms were more pronounced for trabecular myocardium. Finally, cluster 15, with a preferential location diffusively in the outer compact myocardium, was found to be related with cell division. As a consequence, we believe that these cells may be cardiomyoblasts participating in the consolidation of the compact myocardium.

Regarding the clusters associated with atrial cardiomyocytes, the three molecular signatures identified presented distinct locations and molecular functions as suggested by Gene Ontology analysis (**Fig 3B**). While cluster 1 was located mainly in the periphery of the atria and presented GO characteristics of appendage formation, cluster 2 partly had a more central location, presenting GO characteristics of cardiac conduction. However, cluster 7 was found to be the most distinct molecular signature both spatially, being localized in the cranial and caudal part of the atria, and molecularly, presenting GO characteristics of morphogenesis and epithelial to mesenchymal transition (**Fig 3B**). These spatial and GO characteristics suggest that these cells might be involved in keeping with the formation of the atrial septum that occurs at this stage of development.

Finally, two different clusters were found to be associated with endocardium-related cells, one being located in the atria and the other one in the ventricles (**Fig 3B**). The cells belonging to these clusters show very distinct spatial localization. However, unlike cardiomyocytes, we did not find notable differences in their gene expression; both present enrichment in GO terms involved in cardiovascular morphogenesis and development (**Fig 3A and 3B**). In addition, DEG between both clusters were found solely expressed in atrial and ventricular cardiomyocytes respectively, suggesting that both clusters identified the same cell type (endocardial cells) in two different spatial contexts (**S6 Fig**).

### An atria-specific subepicardial signature during human heart development

Perhaps one of the most remarkable aspects of the analysis was the identification of very thin sub-epicardial mesenchymal cell layers in the time point-specific analysis of PCW 6.5, possibly originating from epithelium via epithelial–mesenchymal transition (EMT) [16,17]. Using pseudotime analysis, we identified two main branches emerging from epicardial cells (cluster 18), which could be indicating two differentiation trajectories: one describing the differentiation of epicardial cells into epicardial derived cells and fibroblasts (i.e., cluster 18-21-26-9-12-10), and a second one describing the differentiation of epicardial cells into atrial cardiomyocytes (i.e., cluster 18-11-1-2-7) (**Fig 4A–4C**). An additional branch connects epicardial derived cells to atrial cardiomyocytes (i.e., cluster 12-27-7-2-1), suggesting that epicardium-derived cells (EPDCs) undergo mesenchymal transition and differentiate into cardiomyocytes [18,19]. By mapping both the spage2vec identities and the pseudotime scores of these branches into the tissue we observed that pseudotime has a clear spatial component, matching with the gradient from the periphery to the interior of the heart in the developing atria (**Fig 4D**). GO analysis of clusters presenting enough supporting scRNA-seq cells show terms enriched for EMT and atrial morphogenesis (**Fig 4E**).

## Discussion

The improvement of targeted spatially resolved transcriptomic approaches [20,21] in terms of signal-to-noise ratio, sequencing depth, number of genes and number of cells analyzed is

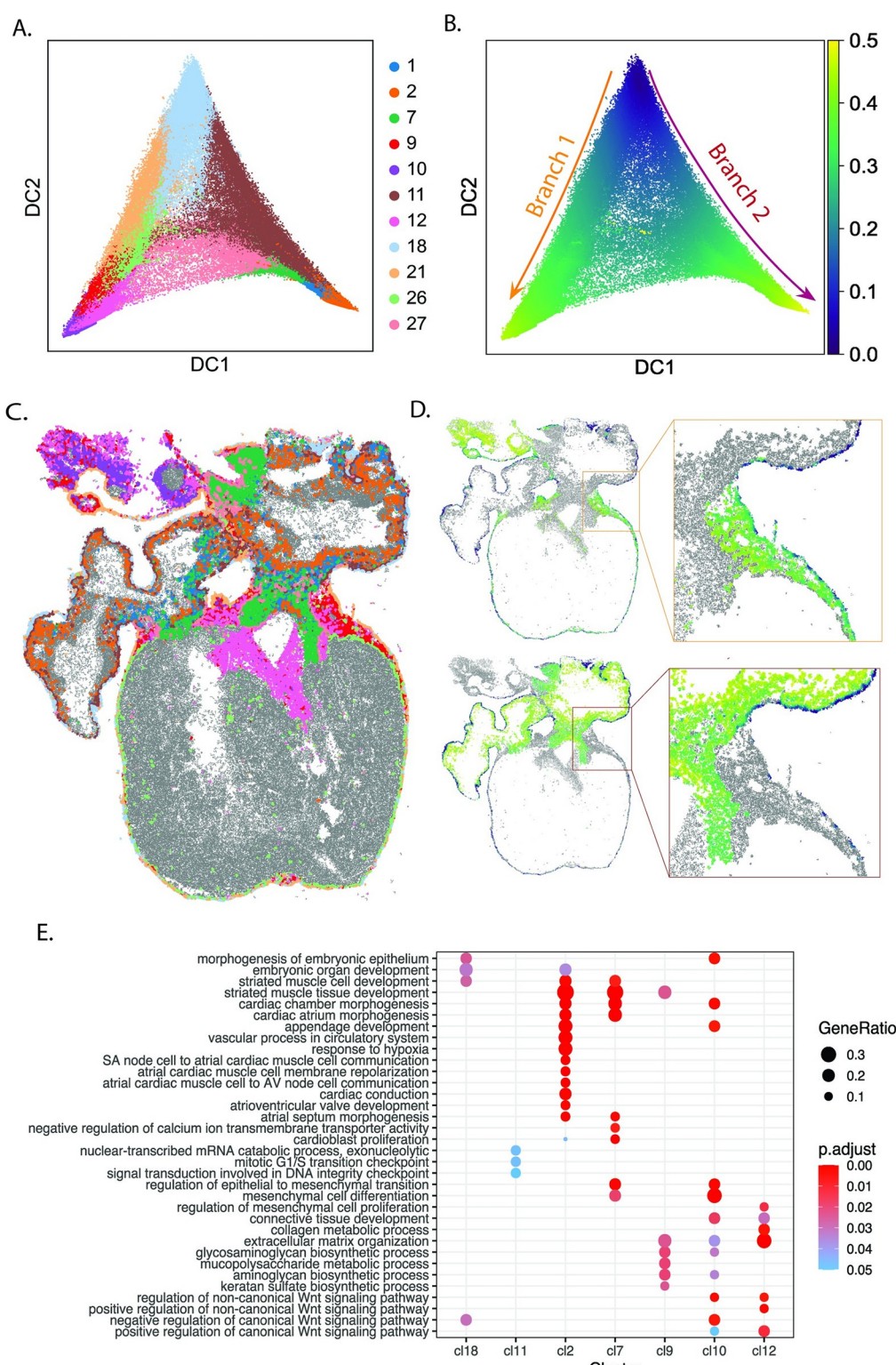

**Fig 4. Description of the differentiation of epicardial cells in the human heart development. A,B.** Diffusion map of pseudo-cell expression profiles defined in PCW 6.5 (Materials and Methods) and assigned to clusters related with epicardial cells, ventricular cardiomyocytes, epicardium-derived cells and fibroblasts. Each spot is labelled in **A** according to the cluster it was assigned to in Fig 2A. In **B**, the color of each spot represents its pseudotime score, considering the root in cluster 18 (epicardial cells). Two main branches can be observed. Pseudotime scores above 0.5

were trimmed for visualization purposes **C.** Spatial map highlighting the spots assigned to the clusters present in Fig 4A in one of the two sections from PCW 6.5. **D.** Spatial map representing the pseudotime scores of each of the spots described in Fig 4B in one of the sections from PCW 6.5 and a region of interest present in the same tissue. Clusters were represented in two different plots, depending on whether they were situated in Branch 1 (top) or Branch 2 (bottom) according to Fig 4A and 4B. **E.** Dot plot showing enrichment of Gene Ontology biological processes for top 15 most differentially expressed genes in the clusters represented in Fig 4A.

leading towards the generation of larger datasets that will enable more and more comprehensive data driven spatial analysis. So far, methods such as ISS have primarily been a useful complement to scRNA-seq strategies by uncovering the spatial location of scRNA-seq defined cell populations. However, spatial molecular organization in itself presents intrinsic critical information of the cellular heterogeneity that is not fully captured by non-spatial methods, thus *de novo* approaches that do not rely on previous knowledge are starting to gain relevance in the field due to their notable advantages [13,22]. In this study, thanks to one of these *de novo* approaches, spage2vec [13], we have been able to define 27 molecular signatures conserved during the developmental process of the heart based solely on the spatial location of the expressed molecules of 69 targeted genes.

In contrast with the original study [7], where cell typing was constrained by availability of scRNA-seq data, our approach is able to define, in a spatiotemporal manner, different molecular signatures stable over the different time points analyzed during the heart development. Our analysis was able to capture stable cell populations through the developmental process, such as epicardial cells, and could be used for understanding biological processes like migration and differentiation. Supervised cell typing approaches [10,12,23] will force the ISS data to fit signatures designed from scRNA-seq, with the risk of introducing biases and losing part of the potential biological information available in the ISS data. Furthermore, supervised approaches may fail to assign cells to a cell type due to discrepancies between the detected molecular signatures and the scRNA-seq data. As a consequence, while *de novo* approaches such as spage2vec assign a molecular signature to each read analyzed, probabilistic cell typing approaches avoid assigning a signature to many of the reads analyzed, missing in some cases molecular patterns with a true biological implication.

Moreover, unlike most existing cell typing strategies, spage2vec does not rely on cell segmentation. This aspect can be highly beneficial when working with compact tissue, where cell borders are difficult to define. Spage2vec directly clusters the mRNA reads based on their local environment, and neighborhood information is incorporated in the process. In order to capture spatial signatures at cellular resolution, this method aggregates local information from neighborhoods within a radius of approximately 15 μm, which is a reasonable inter-cell distance, although the detected spatial clusters can represent cellular and even subcellular gene expression patterns. Since the method is completely unsupervised, super-cellular or sub-cellular patterns may also be captured depending on multiple factors that are related to the gene panel selected, sequencing resolution, and local differences in cell density. This needs to be taken into consideration when interpreting the outcome of the method since individual cell types present in different spatial contexts can be split in different environment-dependent clusters due to the effect of the neighboring cells in the creation of the embedding, as we believe it occurs with endocardium-related cells in this study. In these cases, further verification of the clusters using independent techniques is recommended.

For its unsupervised analysis, spage2vec depends on a targeted ISS gene panel. In this case, the genes were selected at an early stage of the Asp work[10], based on scRNA-seq and Spatial Transcriptomics data. Despite the clear limitation of using a subset of markers for identifying clusters *de novo*, we have shown that leveraging deep learning representation power, spage2vec

can also identify subpopulations through non-linear aggregation of spatial marker features, even without marker genes that can directly identify all cell populations. This is exemplified when comparing atrial and ventricular clusters described here with those reported in a follow up study by Sylvén et al [24] based on the combination of scRNA-seq and Spatial transcriptomics. They report two atrial clusters, trabecular and conduit atrium, that annotate to similar tissue distributions as spage2vec clusters pcw6-1-2 and pcw6-7. While available gene markers annotate trabecular atrium in spage2vec, no highly specific markers are available for conduit atrium but rather a concerted profile of markers that identifies it.

For the ventricular clusters, similar principles prevail. Thus, Sylvén et al. [24] report characteristics of compact, trabecular with subtypes, and Purkinje-related myocardium that have tissue distributions similar to spage2vec clusters pcw6-4,8,15 (see viewer) annotating outer and inner ventricular myocardium and thus with spage2vec based on concerted gene profiles. An exception is spage2vec cluster 24 with CDK1, NUSAP1 and TOP2A gene expressions that are highly specific for cardiomyoblasts with high fractions of cell cycle G2M and S phases and exosome-enriched gene expressions.

Apart from its ability to capture specific subpopulations, here we prove that segmentation-free methods can be used to describe a differentiation process, including its spatial component. In this manuscript we report two main trajectories involving epicardial cells in atrial development. In fact, this observation is supported by Singh et al. 2013[25], Greulich et al. 2011[26], Cai et al. 2008[18] and Zhou et al. 2008[19], who report that at the atrial level epicardial cells flow into the atrial myocardial wall of venous origin and through epithelial-to-mesenchymal transition differentiate into arterial endothelium, smooth muscle and perivascular fibroblasts and may contribute to myocardialization of the atrial wall.

All in all, by applying spage2vec to study the human heart development we have been able to perform a spatiotemporal analysis of the cells found in post conception week 4.5–5, 6.5 and 9, identifying different molecular signatures and developmental processed previously undescribed with this resolution. The spatial maps of the newly characterized cell-type signatures are made public available and can be interactively explored at https://tissuumaps.research.it. uu.se/human_heart.html. Furthermore, this study demonstrates the advantages of using de novo strategies to jointly model the spatial gene expression at different developmental stages, without relying on cell segmentation and scRNA-seq to characterize developmental processes. Thus, it opens the possibility of applying this technique to similar biological systems where reference single cell RNA sequencing data may be limited or not available.

## Material and methods

### Datasets

The ISS dataset of the developing human heart [7] comprises gene expression information of 69 marker genes and decoded spatial coordinates of mRNA spots in eight tissue sections at three developmental time points (Fig 1A). There are 189541, 812808, and 1471602 mRNA reads at the three time points respectively, summing up to a total of 2473951 reads.

### Spatiotemporal representation of ISS gene expression data with spage2vec

Spage2vec [13] learns to map local neighborhood relationships between mRNA spots as distances in a continuous latent space using a deep learning model. As a result, a numerical vector is assigned to each individual mRNA spot describing its neighborhood composition. Therefore, molecules that share similar local environments are described with numerically similar vectors and consequently mapped in close proximity in the learned latent space. In such a way, we are able to build a spatiotemporal representation of the spatial gene expression in an

unsupervised manner and without using any prior information. The learned representation is then used to perform clustering analysis to define localized gene expression signatures that represent cell-type signatures across the three embryonic stages.

## Constructing a spatial gene expression graph

We first construct an undirected graph where each node represents an mRNA spot, with a one-hot encoding feature vector representing its corresponding gene. Each node is then connected by edges to its spatial local neighbors of the same tissue section within a maximum distance (d_max = 44.9 pixels/ 14.58 μm). We estimate the maximum distance such that 99% of nodes in the graph are connected to at least one neighbor. Connected components with less than six nodes are successively removed from the graph to exclude spurious reads such as spots located outside of the region of the heart sample, thus leaving 97.7% of the original mRNA reads for further processing.

## Graph convolutional neural network model and training

We then train a graph convolutional neural network on the spatial gene expression graph to produce the spage2vec latent representation for each mRNA spot. The neural network consists of two GraphSAGE [27] layers. At each layer, the features of a node and its local neighborhood are aggregated and propagated to the next layer. The neural network learns its parameters in an unsupervised setting by minimizing a loss function based on random walks. The loss function of a node encourages similarity between the node and a direct neighbor that occurs in a random walk, and dissimilarity between the node and another node randomly sampled from the graph. Regarding the hyperparameters of the model, we use the mean aggregator at each layer and ReLU as the activation function for the first layer. The size of each layer is 32. The model is trained for 10 epochs with a batch size equal to 64, using Adam optimizer [28] with a learning rate equal to 0.001. The output for each mRNA spot is then a spage2vec latent vector of length 32.

## Cluster analysis and visualization

After predicting a latent vector for each mRNA spot based on its neighborhood composition, we compute a kNN (k = 15) weighted graph of the spage2vec latent vectors and apply the Leiden clustering algorithm [29] (with clustering resolution r = 1) on the kNN-graph. We then use PAGA [30] to quantify the connectivity of acquired clusters, representing the clusters' proximity in the latent space. Each cluster with less than 1000 nodes is merged into the closer larger cluster in the PAGA graph having the maximum connectivity to the smaller cluster, if the connectivity was greater than 0.1. Otherwise, they are considered outliers and filtered out. After merging and filtering out the small clusters, we count the number of spots per cluster per gene followed by cluster-wise Z-score normalization to create a cluster expression matrix. This led to the final set of spage2vec clusters, which can be visualized interactively using TissUU-maps [14].

## Spage2vec and scRNA-seq data integration

We perform data integration between spage2vec clusters of individual analysis of PCW 6.5 ISS data and the corresponding scRNA-seq data from Asp et al. Specifically, we first log-normalize scRNA-seq total counts per cell. Then, we generate pseudo-cell gene expression profiles for each mRNA spot by aggregating its k-nearest neighbor (k = 100) in the spage2vec latent space. Next, we filter genes with less than 100 reads and log-normalize total counts per pseudo-cell.

We thereafter integrate pseudo-cell and scRNA-seq gene expression profiles using SpaGE [15]. The two datasets are aligned by projecting them in a common latent space by domain adaptation [31] using 30 principal vectors. After alignment, we can either infer the spatial profile of genes that are missing from the original ISS gene panel, or vice versa assign scRNA-seq cells to spage2vec clusters by k-nearest neighbor imputation.

Specifically, for each scRNA-seq cell we compute a cosine similarity in the common latent space with respect to all the k-th (k = 15) nearest neighbor pseudo-cells, and we define correspondence with a spage2vec cluster as the sum of all cosine similarities with respect to those pseudo-cells belonging to the given cluster. We then exclude scRNA-seq cells with low correspondence to the spatial clusters (i.e. maximum cosine similarity smaller than 0.3), and we assign each scRNA-seq cell to the spage2vec cluster with highest cosine similarity. Spatial clusters with less than 10 scRNA-seq cells assigned are marked as weakly aligned as they miss enough supporting scRNA-seq cells and thus are excluded from further analyses.

## Supporting information

**S1 Fig. Comparison of spage2vec clusters with cell-type annotations from Asp et al. A**. Heatmap representing the confusion matrix between the cell type assigned to each read via pciSeq in Asp et al. [7] and the spage2vec cluster annotations. **B**. Heatmap representing the correlation between the expression profile of each spage2vec cluster and each cell type described using scRNA-seq in Asp et al. [7] for the 69 genes included in both datasets. **C**. Spatial location of a subset of clusters from the spage2vec analysis (top) and pciSeq (bottom) in a specific sample from pcw 6.5. Clusters selected represent both epicardial cells and fibroblast-like cells /epicardium derived cells in both cases and colors have been based on the similarities between spage2vec clusters and pciSeq clusters. A zoomed in region is shown for both datasets. **D**. Spatial location of cardiomyocyte-related clusters defined by both spage2vec (top) and pci-seq (bottom) in a specific section from pcw 6.5. Each spage2vec cluster assigned to cardiomyocytes were classified as atrial or ventricular according to their molecular signature in Fig 1C. (TIF)

**S2 Fig. Correspondence between pciSeq and spage2vec clusters. A.** Map of the main morphological regions identified in pcw6.5 sections. Regions were calculated by redefining every read based on the reads present in a radius of 70 pixels/22.8 um to capture the main tissue domains and applying leiden clustering on it. **B.** Scatter plot representing the abundancies of specific paired spage2vec-pciseq clusters in the regions defined in S2A (from left to right: epicardial cells-cluster 20; capillary endotheium-cluster 18; cardiac neural crest cells-cluster 26 and Fibroblast-like (VD)-cluster 16). Pearson correlation for every pair of clusters is included in the scatter plot. (TIF)

**S3 Fig. Integration of time point-specific analyses. A.** PAGA plot representing all clusters found in the time-point specific analyses of pcw 4.5–5, pcw 6.5 and pcw 9. Each cluster is represented in a node and backg//round colors indicate main cell type annotations. **B.** Spatial location's comparison between general clusters (Figs 1, S1, S2 and S3) and time-point specific clusters. Three main clusters are represented: cluster 13 (left), cluster 20 (middle) and cluster 18 (right) in one of the samples of each time point, together with their correspondent time point-specific cluster. (TIF)

**S4 Fig. Correspondence between general clusters and time point-specific cluster.** Heatmap representing the confusion matrix between the cluster assigned to each read in the general

analysis (Figs 1, S1, S2 and S3) and the cluster assigned to each read in the time point-specific analysis. Color column situated next to the time-point specific cluster labels indicates the time point where each cluster has been detected. The color code used in Fig 1A is used to label each time point.
(TIF)

**S5 Fig. Differential expression analysis of scRNA-seq data based on spage2vec cluster annotations.** Top 15 differentially expressed genes for the clusters found in the individual analysis of pcw 6.5. Scores of each gene (y axis) corresponds to the Wilcoxon rank-sum test score.
(TIF)

**S6 Fig. Exploration of differentially expressed genes between the endothelial and cardiomyocyte-related clusters. A**. Heatmap representing the correlation patterns found in scRNA-seq (left) and spage2vec (right) between the three most differentially expressed genes (DEG) between endothelium-related clusters represented in Fig 3A. **B.** Dot plot representing the expression detected via scRNAseq of the three most DEG between endothelium-related clusters in the cell types identified in Asp et al. **C.** Heatmap representing the correlation patterns found in scRNAseq (left) and spage2vec (right) between the three most DEG between atrium cardiomyocytes subclusters represented in Fig 3A. **D.** Dot plot representing the expression detected via scRNAseq of the three most DEG between atrium cardiomyocytes subclusters in the cell types identified in Asp et al. **E.** Heatmap representing the correlation patterns found in scRNAseq (left) and spage2vec (right) between the three most DEG between ventricular cardiomyocytes subclusters represented in Fig 3A. **F.** Dot plot representing the expression detected via scRNAseq of the three most DEG between ventricular cardiomyocytes subclusters in the cell types identified in Asp et al. **G.** Dot plot representing the expression of the 2 most DEG of each fibroblast/EPDC subcluster (cl) **H.** Dot plot representing the expression detected via scRNAseq of the top DEG between fibroblast/EPDC subclusters in the cell types identified in Asp et al.
(TIF)

**S7 Fig. Exploration of spage2vec pcw6.5 clusters. A.** Heat map representin the mean expression of each spage2vec pcw6.5 cluster for the genes included in the ISS panel. **B.** Heat map representing the cross correlation of the spage2vec pcw6.5 clusters based on their mean expression. **C.** Jaccard index derived from the integration of the scRNA-seq dataset and the spage2vec pcw6.5 clusters **D.** Dot plot representing the expression of the genes expressed in the spage2vec pcw6.5–13 cluster in the different scRNA-seq cell types. **E.** Region of interest of the *in situ* sequencing samples (pcw6.5) representing the location of cluster pcw6.5–13 (right) and some of genes expressed in this cluster (left) **F.** Dot plot representing the expression of the genes expressed in the spage2vec pcw6.5–16 cluster in the different scRNA-seq cell types. **G.** Region of interest of the *in situ* sequencing samples (pcw 6.5) representing the location of cluster pcw6.5–16 (right) and some of genes expressed in this cluster (left)
(TIF)

## Acknowledgments

We thank Fredrik Nysjö and Christophe Avenel at the BioImage Informatics Facility, funded by SciLifeLab, the National Microscopy Infrastructure, NMI (VR-RFI 2019–00217), and the Chan-Zuckerberg Initiative, for help and support on data visualization. We'd also like to thank Christoffer Mattsson Langseth for his contribution in designing the study.

## Author Contributions

**Conceptualization:** Sergio Marco Salas, Mats Nilsson, Carolina Wählby, Gabriele Partel.

**Data curation:** Sergio Marco Salas, Xiao Yuan, Gabriele Partel.

**Formal analysis:** Sergio Marco Salas, Xiao Yuan, Christer Sylven, Gabriele Partel.

**Investigation:** Sergio Marco Salas, Xiao Yuan, Christer Sylven, Gabriele Partel.

**Resources:** Sergio Marco Salas, Xiao Yuan.

**Software:** Sergio Marco Salas, Xiao Yuan, Carolina Wählby, Gabriele Partel.

**Supervision:** Carolina Wählby, Gabriele Partel.

**Visualization:** Sergio Marco Salas, Gabriele Partel.

**Writing – original draft:** Sergio Marco Salas, Christer Sylven, Mats Nilsson, Gabriele Partel.

**Writing – review & editing:** Xiao Yuan, Mats Nilsson, Carolina Wählby, Gabriele Partel.

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
