## [Decision Letter · Decision Letter 0]

11 Mar 2022

Dear Professor Nilsson,

Thank you very much for submitting your manuscript "De novo spatiotemporal modelling of cell-type signatures in the developmental human heart using graph convolutional neural networks" for consideration at PLOS Computational Biology.

As with all papers reviewed by the journal, your manuscript was reviewed by members of the editorial board and by several independent reviewers. In light of the reviews (below this email), we would like to invite the resubmission of a significantly-revised version that takes into account the reviewers' comments.

We cannot make any decision about publication until we have seen the revised manuscript and your response to the reviewers' comments. Your revised manuscript is also likely to be sent to reviewers for further evaluation.

Sincerely,

Qing Nie

Associate Editor

PLOS Computational Biology

Daniel Beard

Deputy Editor

PLOS Computational Biology

Reviewer's Responses to Questions

**Comments to the Authors:**

Reviewer #1: Review Comments:

Graph neural networks has recently become a very widely used and powerful method for analyzing diverse complex networks, e.g., social network, financial network, protein networks, brain networks [1]. The paper applied the Spage2vec method, an advanced graph neural network model, to effectively characterize the cell types at sub-cellular imaging resolution for the development of fetal heart from 4.5 to 9 post conception weeks (PCWs)---three developmental stages. However, the in situ pciSeq technique published in 2020 only explored cell diversity for the 6.5 PCW heart development stage, yet not considered the 4.5-5 PCW and 9 PCW time points.

Spage2vec requires no cell segmentation, and can learns the latent cell expression representations by incorporating the critical spatial molecular organization structure property. Using the obtained latent cell representations, it enables to characterize previously unreported molecular diversity within cardiomyocytes and epicardial cells. Moreover, it can identify their characteristic expression signatures for diverse downstream tasks, e.g., subpopulation detection. The experimental results show that the proposed method enables to discover 27 meaningful cellular clusters (more than the 12 cell types defined in the scRNA-seq data) shared across the three different heart development embryonic stages.

The authors also built an open-source GitHub code repo for their work and provided an online platform for interactively visualization of multiple cell type clusters in different colors. Having public repo and online visualization platform can greatly aid other researchers in this field in better understanding the method implementations and the entire study.

After reading through the paper, I listed some of my comments and questions below.

Page 3, Line 70-71:

If I understood well, the study on the pciSep technique was initially developed for brain cortex, yet not for the development of the heart. However, the author described that “the overall study was the first comprehensive spatial atlas of the developmental human heart”.

Page 7 Line 147:

There is a typo in the sentence “clusters whith (typo) distinct and consistent spatial…”.

Material and Methods:

• Page 13, Line 301/304: Does Spage2vec can capture global graph structure property, i.e., second-order proximity, or just capture local neighborhood relations among mRNA spots?

• Line 311, page 14: I would suggest to add more details about the number of nodes in the constructed graph, and what is the node feature dimension? What is the adjacency matrix dimension in the constructed graph? weighted or unweighted, directed, or undirected?

• What are the performance if directly apply conventional clustering t-sNE or IsoMap

• Is there any correlation between the three different development stages? If has, is it possible to learn these evolving dynamics information directly using spage2vec model? or using Spage2vec to capture both spatial and temporal graph embeddings in the latent space simultaneously?

Reviewer #2: Dear authors,

I have reviewed the manuscript titled “De novo spatiotemporal modelling of cell-type signatures in the developmental human” presented by Salas et al. In their work they present their re-analysis of the human developmental heart spatial data presented by Asp et al 2019 using a graphical modelling approach that is independent of identifying cells. In doing so, they identify previously undescribed molecular diversity within cardiomyocytes and epicardial cell populations and describe gene expression trajectory dynamics within the tissues.

Both the methodolocial approach and results are indeed very interesting, and I would like to see them published. Cell segmentation free approaches are still underappreciated in spatially resolved transcriptomics data analysis.

However, I feel that the paper is currently lacking investigation and validation of the differences between the obtained results, and those presented by Asp et al. I hope that I can elaborate on my concerns with the following points.

Major

1) Comparative analysis of the spage2vec and pciSeq/scRNAseq data seems lacking in places. Some specific points regarding this issue are:

1a) Methodological differences leading to different cluster numbers. In scRNAseq it is well established that the clustering parameters play a major role in how many clusters are identified. How invariant are the 27 clusters to parameters used by spage2vec (line 120)? How invariant are the 12 clusters to parameters used by pciSeq? How invariant are the 15 clusters to parameters used by scRNAseq? The authors should investigate that reasonable parameter changes in e.g. the scRNAseq cluster do not result in e.g. 27 clusters that match the spage2vec clusters perfectly (.. which is unlikely, but it should be explored). If the aut

1b) I would expect that for the 6.5 PCW dataset (where matched scRNAseq and ISS data exist) that the cell types obtained between the spage2vec can be found in the scRNAseq data. The authors postulate that the detection of additional clusters in the ISS data using spage2vec is due to sampling more cells - this would explain the difference for rare cell types, but would also implicate that cell types with many cell should be equally represented in both the scRNAseq and ISS data. However given that such a large proportion of those cells are seen in the spage2vec data and not in the spage scRNAseq analysis makes it unlikely that this arises for differences in cell number. For example, the spage analysis in Figure 2 shows that a number of spage2vec e.g. clusters 16, 24 ,19 , 23 , 13, 25 and 3 do not appear to have matches in the spage analysis of the scRNAseq data (fig 2B). However, nearly all of these spage2vec clusters have have many cells annotated in the tissue (fig s2) at 6.5PWC. In light of this, how do the authors explain this discrepency?

1c) If the authors cannot recapitulate the novel spage2vec cell types in the scRNAseq data, the authors need to explain why this is the case, with the support from other studies or by validation.

1d) I find the consistency of the analysis of the Spage scRNAseq clusters (e.g. presented in figure 3A, S4E, S7) to be confusing, as the scRNAseq cluster variability do not correspond to spage2vec clusters variability. An example of how this can lead to confusion is that in cluster 14 and 17 have very different gene expression (e.g TCIM in cluster 14, and MYL2 for 17) in figure 1C, however, neither of these genes appear in the most variable genes for 14 and 17 in Figure 3A, but instead the genes shown in 3A seem to not be very variable at all. How do the authors explain this?

1e) While spage2vec resolves much more sub-cell-types/state in e.g. atrial cardiomyocytes, it does not capture “Fibroblast-like (AV-rel)”, “Endothelium/ pericytes” (Figure S4A), “Erythrocytes 2”, or “Myoz2-enriched cardiomyocytes” very well. This is partially contradictory to the statement of resolving cell type heterogeneity better than scRNAseq (lines 145-147). The authors should clarify the implications of this on this current study, and critically assess the previous cell-typing in Asp et al. Looking at figure S4D, it seems that Spage2Vec misses the visual enrichment of “Myoz2-enriched cardiomyocytes” in the inner part of the left ventricle (and to lesser extent in the right ventricle). The authors should explain how the spage2vec clusters are an improvement over the pciseq clusters in figure S4D.

1f) How was the spatial concordance of cell types determined (line 127-128)? This should be performed statistically.

1g) The authors postulate that the spage2vec results are more inline with expected distributions of cell types (line 132-133). Please provide evidence for this, especially in light of point 1b.

Intermediate

2. Is spage2vec a “graph convolutional neural network”? While there are elements of graphs, convolutional operations, and neural networks in the implementation, the authors should revisited this to see whether their description fits with the current level of precision in defining tools in the field. Interestingly, the word “convolution” does not appear in the original spage2vec publication.

3. The methods are well described, but lack any information of computational frameworks/tools/versions that are used. For example, the authors do not describe whether their analysis is performed in matlab/R/Python/etc for numerous sections. In fairness, the authors provide access to a very good Jupyter notebook in their github repository, but (i) the link to the repository it not a DOI (see https://docs.github.com/en/repositories/archiving-a-github-repository/referencing-and-citing-content), and (ii) the authors and editors should decide on the level of appropriate reporting of computational tools (e.g. for NPG, I would typically list the version of tools and libraries within the methods text)

4. It would be interesting to see how spage2vec compares to results obtained by HMRF based methods (e.g. https://dx.doi.org/10.1038%2Fnbt.4260, https://doi.org/10.1038/s41587-021-01044-w) for this dataset. While this would be interesting, a systematic comparison this may fall out of scope of the current study, so perhaps the authors could comment on how they expect their method would hold up against these other models.

5. The human heart contain multi-nucleated cells, that may not be amenable to microfluidics based scRNAseq. Can the authors comment on the extent to which they expect this to be an issue in the scRNAseq data analysis, and how this may potentially link to cardiomyocyte diversity identified by spage2vec?

6. Why do the authors only find “immune” cells in the 6.5PCW dataset?

7. Datasets. There isn’t a link to the datasets that were analysed in the paper. On the github repo, the link to the raw data seems broken: https://doi.org/10.5281/zenodo.5060858.

Minor (these may be suggestions or questions):

8. Subheadings in the results would make is easier to read..

9. As a general minor issue, the paper reads like “I used tool X on dataset Y”, with spage2vec being mentioned a few too many times (it sometimes appears three times within a paragraph)

10. The abstract could perhaps also mention their trajectory analysis results.

11. Line 50. “Cell Atlasing” -> “cell atlasing”.

12. Line 55. Should “Single cell” be “Single cell RNAseq”?

13. Line 56. Are citations 5-9 appropiate? Given the large number of SRT methods, perhaps one or 2 review papers should be cited?

14. Line 68-69. I cannot see what this states in addition to lines 67-68.

15. Line 70. The use of “comprehensive” together with “spatial atlas” here is questionable (… to be clear Asp et al was a great study, but we should be precise with what the results were). Are the authors sure that there were no other (non-transcriptomics) imaging based analysis of the human heart? Likewise, while in the “spatial atlas” is synonymous with “spatial transcriptional atlas” in many field of science, in other fields it could mean something different (e.g. “spatial proteomic atlas”, or atlasing using radiological approaches). The authors should either tone down “comprehensive”, or rephrase the result (e.g. “spatial transcriptional atlas”/”spatial cellular atlas”).

16. E.g. Line 85. Should “reads” be replaced for something more generic, to also be applicable for, e.g. FISH based molecule detection? The nature of this problem is not specific to ISS, but to all single molecule resolved SRT assays.

17. Line 91. “Powerful” – is this hyperbole? Graph-based approaches are indeed useful, but the use of “powerful” does not add to the sentence.

18. Line 149. The differences in the number of cells in the ISS and scRNAseq data should be enumerated.

19. Line 383. “e.i.” should be “i.e.”

Figures

20. Please check that figures are shown consistently. E.g Figure S4 shows the 6.5 PWC image, but this is flipped compared figure 1D.

21. Figure 1C. The scale bar has no units.

22. Figure 2B. The scale bar has no units. The x-axis label should be improved, e.g. “PCW6.5 spage2vec clusters”.

23. Figure 3A. Abbreviation “cl” is not explained. Perhaps add “Cluster” as the top axis label?

24. Figure 3B. X-axis labelling is inconsistent.

25. Figure S1, S2, S3. It would be nice if the panels were arranged by biological grouping.

26. Figure S5. Perhaps this should be a main figure.

27. Figure S6. No units on the scale bar. X-axis label should be “spage2vec cluster”

**Have the authors made all data and (if applicable) computational code underlying the findings in their manuscript fully available?**

Reviewer #1: Yes

Reviewer #2: **No: **The zenodo data repository link is provided, but the link does not work.

PLOS authors have the option to publish the peer review history of their article (what does this mean?). If published, this will include your full peer review and any attached files.

Reviewer #1: No

Reviewer #2: No
---

## [Decision Letter · Decision Letter 1]

6 Jul 2022

Dear Professor Nilsson,

We are pleased to inform you that your manuscript 'De novo spatiotemporal modelling of cell-type signatures in the developmental human heart using graph convolutional neural networks' has been provisionally accepted for publication in PLOS Computational Biology.

Best regards,

Qing Nie

Associate Editor

PLOS Computational Biology

Daniel Beard

Deputy Editor

PLOS Computational Biology

Reviewer's Responses to Questions

**Comments to the Authors:**

Reviewer #2: Dear authors,

I have reviewed the point by point response and resubmitted work by Salas and colleagues. I am satisfied with the responses they provide and would recommend the work for publication.

Thank you for entertaining my criticisms, and congratulations on a nice study.

**Have the authors made all data and (if applicable) computational code underlying the findings in their manuscript fully available?**

Reviewer #2: Yes

PLOS authors have the option to publish the peer review history of their article (what does this mean?). If published, this will include your full peer review and any attached files.

Reviewer #2: **Yes: **Naveed Ishaque

---

## [Editor Report · Acceptance letter]

23 Jul 2022

PCOMPBIOL-D-22-00047R1 

De novo spatiotemporal modelling of cell-type signatures in the developmental human heart using graph convolutional neural networks

Dear Dr Nilsson,

I am pleased to inform you that your manuscript has been formally accepted for publication in PLOS Computational Biology. Your manuscript is now with our production department and you will be notified of the publication date in due course.

With kind regards,

Zsofi Zombor
